# DS-VIC: Unsupervised Discovery of Decision States for Transfer in RL

## Abstract

We learn to identify 'decision states', namely the parsimonious set of states where decisions meaningfully affect the future states an agent can reach in an environment. We utilize the VIC framework (Gregor et al., 2016), which maximizes an agent's 'empowerment', *i.e.* the ability to reliably reach a diverse set of states – and formulate a sandwich bound on the empowerment objective that allows identification of decision states. Unlike previous work (Goyal et al., 2019), our decision states are discovered without extrinsic rewards – simply by interacting with the world. Our results show that our decision states are: 1) often interpretable, and 2) lead to better exploration on downstream goal-driven tasks in partially observable environments.

## 1 Introduction

Not all states in a decision making process are created equal. Consider the middle illustration in Figure 1, where a robot has four 'options', each representing a goal shape it can potentially reach. Given its spawn location, it can initially proceed straight regardless of the option it chooses until the intersection, at which point it needs to utilise 1 bit of (Shannon) information from the option variable to inform the choice of whether to turn left or right. However, right after it takes the left turn (say), it again does not need the goal information when choosing actions and can follow 'default' or 'option-independent' behaviour. Thus, there is a natural difference in the minimum amount of necessary goal information or 'relevant information' (Polani et al., 2006) needed at different states.

Identifying these 'decision states', *i.e.* states in the environment where a high amount of relevant goal information is needed, leads to better understanding of the environment structure, which has the potential to enable better transfer to novel environments and tasks. This has previously been shown by Goyal et al. (2019) in partially-observable, goal-driven (*i.e.* with extrinsic rewards) 2D navigation settings, where using decision states to guide exploration enables faster learning.

**Goal-independent Decision States**. Our key intuition is that decision states exist in an environment independent of extrinsic goals. For example, one can imagine the (middle) red tile at the intersection in Figure 1 being a useful decision state – even across all possible navigation goals the agent could have. Ofcourse, not all intersections are decision states. For *e.g.*, if the left room was full of lava, where the robot would die, thinking of the middle red tile as a decision state would have limited utility, since the relevant goal information (across all reachable goals) in this case would be 0 bits.

Identifying 'unsupervised' (or task-agnostic) decision states is advantageous in scenarios where: 1) rewards are sparse or absent (Pathak et al., 2017), 2) for an agent to learn meaningful behaviour, proxy goals and rewards need to be hand engineered (making it hard to scale), and 3) the notion of a goal might not even be obvious, *e.g.* in continuous control tasks (Lillicrap et al., 2015).

Our DS-VIC method identifies decision states without extrinsic rewards, solely by interacting with partially-observable environments. These decision states are a function of the environment as well as the states an agent can reliably reach. We demonstrate that our decision states generalise to novel environments and tasks, leading to comparable (or better) exploration and performance on downstream tasks compared to Goyal et al. (2019) that identifies goal-driven decision states.

**Decision States in Empowered Agents.** We build on the VIC framework (Gregor et al., 2016) (Figure 1), where the core idea is to maximize the *empowerment* of an agent. In simple terms, an empowered agent is able to reliably reach a diverse set of states in the environment, while avoiding states that are difficult to reach reliably. We use VIC's formulation to learn options $\Omega$ that maximize (a lower bound $\mathcal{J}_{VIC}$ on) the mutual information $I(S_f, \Omega)$ where $S_f$ is the final state in a trajectory. We use latent options since we operate in a setting without external goals. To see why this maximizes

Figure 1: Left: The VIC framework (Gregor et al., 2016) in a navigation context: an agent learns high-level macro-actions (or options) to reach different states in an environment reliably without any extrinsic reward. Right: DS-VIC identifies decision states (in red) associated with options where the agent is empowered to make informed decisions. The rest of the states in a trajectory (dotted lines in blue regions) then show default, option-independent behavior. Identification of decision states leads to improved transfer to novel environments.

empowerment, notice that $I(S_f, \Omega) = H(S_f) - H(S_f|\Omega)$, where $H(.)$ denotes entropy. Thus, empowerment maximizes the diversity in final states $S_f$ while learning options highly predictive of $S_f$ (Salge et al., 2013). Using VIC, we identify decision states by attempting to hit the relevant information required at a particular state by minimizing the mutual information $I(\Omega, A_t|S_t)$ (for every state $S_t$, action $A_t$). This is similar in spirit to Goyal et al. (2019); Polani et al. (2006), with the key difference that we use latent options $\Omega$ instead of external goals. Interestingly, we prove that $\sum_t I(\Omega, A_t|S_t)$ is an upper bound on empowerment, which we minimize, in addition to maximizing the VIC lower bound $\mathcal{J}_{VIC}$. This shines a connection to VIC– while they optimize a lower bound, our work optimizes a sandwich bound on the empowerment[1]. To summarize our contributions,

- We propose DS-VIC, a novel framework to identify decision states in a task-agnostic manner. These decision states align with our intuitive assessments in various partially-observed scenarios.
- We provide an understanding of our proposed objective, proving that our framework optimizes a sandwich bound on the empowerment $I(\Omega, S_f)$.
- We show that our mechanism to identify decision states is transferable and leads to improved sample-efficiency on goal-driven tasks in novel environments. On a challenging grid-world navigation task, our method outperforms (a re-implementation of) Goyal et al. (2019).

## 2 METHODS

### 2.1 NOTATION

We consider a Partially Observable Markov Decision Process (POMDP), defined by the tuple $(\mathcal{S}, \mathcal{A}, \mathcal{P}, r)$, where the (discrete or continuous) state-space $\mathcal{S}$ is a set of underlying but unobserved states $s \in \mathcal{S}$ and $\mathcal{A}$ denotes a discrete action space. $\mathcal{P} : \mathcal{S} \times \mathcal{S} \times \mathcal{A}$ denotes an unknown transition function, representing $p(s_{t+1}|s_t, a_t) : s_t, s_{t+1} \in \mathcal{S}, A_t \in \mathcal{A}$. Both VIC and DS-VIC train an option ($\Omega$) conditioned policy $\pi(a_t|\omega, s_t)$, where $\omega \in \{1, \cdots, |\Omega|\}$. Following standard practice (Cover & Thomas, 1991), we denote random variables in uppercase ($\Omega$), and items from the sample space of random variables in lowercase ($\omega$).

### 2.2 VARIATIONAL INTRINSIC CONTROL (VIC)

The option $\Omega$ in VIC is a global latent variable underlying a trajectory $\boldsymbol{\tau} = \{S_0, A_0, \cdots, S_f\}$ (Figure 2). To encourage the agent to reach a diverse set of states reliably, VIC proposes maximizing the mutual information (Cover & Thomas, 1991) between $\Omega$ and final state $S_f$ given $s_0$, *i.e.* $I(S_f, \Omega \mid S_0 = s_0)$. Informally, this maximizes the empowerment for an agent, *i.e.* its internal options $\Omega$ have a high degree of correspondence to the states of the world $S_f$ that it can reach. VIC formulates a variational lower bound on this mutual information. Specifically, let $p(\omega \mid s_0) = p(\omega)$ be a prior on options (we keep the prior fixed as per Eysenbach et al. (2018)), $p^J(s_f \mid \omega, s_0)$ is defined as the (unknown) terminal state distribution achieved when executing the policy $\pi(a_t \mid \omega, s_t)$, and $q_\nu(\omega \mid s_f, s_0)$ denote a (parameterized) variational approximation to the true posterior on options

---

[1]This has potential implications beyond the context of this paper, for example, in training VIC models which target a particular amount of "compression" in the option ($\Omega$) representation, similar representational choices for supervised (Tishby et al., 1999) and unsupervised learning (Alemi et al., 2016).

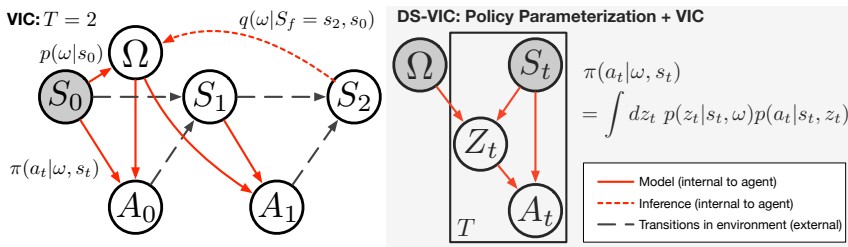

Figure 2: **Illustration of VIC for 2 timesteps. L:** Given a start state $S_0$, VIC samples option $\omega$ and follows policy $\pi(a_t \mid \Omega = \omega, s_t)$ and infers $\Omega$ from the terminating state ($S_2$), optimizing a lower bound on $I(S_2, \Omega \mid S_0)$. **R:** DS-VIC considers a particular parameterization of $\pi$ and imposes a bottleneck on $I(A_t, \Omega|S_t)$.

given $S_f$ and $S_0$. Then:

$$I(\Omega, S_f \mid S_0 = s_0) \geq \mathbb{E}_{\Omega \sim p(\omega), S_f \sim p^J(s_f | \Omega, S_0 = s_0)} \left[ \log \frac{q_\nu(\Omega \mid S_f, S_0 = s_0)}{p(\Omega)} \right] = \mathcal{J}_{VIC}(\Omega, S_f; s_0) \tag{1}$$

## 2.3 DECISON STATE VIC (DS-VIC)

We identify decision states by employing an information regularizer that finds states with minimum relevant option information. Formally, this means that at every timestep $t$ in the trajectory, we minimize the mutual information $I(\Omega, A_t|S_t, S_0 = s)$ and the resulting states where this mutual information remains high despite the minimization are identified as decision states. Intuitively, this means that on average (across different options), a decision state is a state with higher relevant option information than other states (*e.g.* the red regions in Figure 1) Overall, our objective is to maximize:

$$\mathcal{J}_{VIC}(\Omega, S_f; s_0) - \beta \sum_t I(\Omega, A_t \mid S_t, S_0 = s_0) \tag{2}$$

where $\beta > 0$ is a trade-off parameter. Thus, this is saying that one wants options $\Omega$ which allow the agent to have a high empowerment, while utilizing the least relevant information at each step.

Interestingly, Equation 2 has a clear, principled interpretation in terms of the empowerment $I(\Omega, S_f|S_0)$ from the VIC model. We state the following lemma (full proof in Appendix):

**Lemma 2.1** *Let $A_t$ be the action random variable at timestep $t$ and state $S_t$ following an option-conditioned policy $\pi(a_t|s_t, \omega)$. Then, $I(\Omega, A_t|S_t, S_0)$ i.e. the conditional mutual information between the option $\Omega$ and action $A_t$ when summed over all timesteps in the trajectory, upper bounds the conditional mutual information $I(\Omega, S_f|S_0)$ between $\Omega$ and the final state $S_f$ – namely the empowerment as defined by Gregor et al. (2016):*

$$I(\Omega, S_f|S_0) \leq \sum_{t=1}^{f} I(\Omega, A_t|S_t, S_0) = \mathcal{U}_{DS}(\boldsymbol{\tau}, \Omega, S_0) \tag{3}$$

**Implications**: With this lens, one can view the optimization problem in Equation 2 as a Lagrangian relaxation of the following constrained optimization problem:

$$\max \mathcal{J}_{VIC} \; s.t. \, \mathcal{U}_{DS} \leq R \tag{4}$$

where $R > 0$ is a constant. Thus, we can essentially target values of $R$ we have in mind by maximizing the lower bound and minimizing the upper bound, by (in practice) sweeping different values of $\beta$ in eq. (2). This is a result potentially of interest beyond the specific context of decision states (Achiam et al., 2018; Gregor et al., 2016). Specifically, while the VIC objective only allows us to maximize the empowerment $I(\Omega, S_f) = H(S_f) - H(S_f|\Omega)$, imposing an upper bound can help control how 'abstract' the latent option representation is relative to the states $S_f$, by finding solutions that have say a higher entropy $H(S_f|\Omega)$ on states given options. This might lead to more abstract options that achieve better generalisation depending on downstream tasks. Note that most approaches currently limit the abstraction by constraining the number of discrete options, which (usually) imposes a (tighter) upper bound on $H(\Omega)$, since $H \geq 0$ in the discrete case. However, this does not hold for the continuous case, where the bound might be more useful. Investigating this is beyond the scope of this current paper, however, as our central aim is to identify decision states, and not to scale the VIC framework to continuous options.

## 2.4 ALGORITHMIC DETAILS

**Upper Bounds for** $I(\Omega, A_t \mid S_t, S_0)$**.** Inspired by InfoBot (Goyal et al., 2019), we bottleneck the information in a statistic $Z_t$ of the state $S_t$ and option $\Omega$ used to parameterize the policy $\pi(A_t \mid \Omega, S_t)$ (fig. 2 right). This is justified by the the data-processing inequality (DPI) (Cover & Thomas, 1991) for the markov chain $\Omega, S_t \leftrightarrow Z_t \leftrightarrow A_t$, which implies $I(\Omega, A_t \mid S_t, S_0) \leq I(\Omega, Z_t \mid S_t, S_0)$. We can then obtain the following upper bound on $I(\Omega, Z_t \mid S_t, S_0)$ (see appendix for derivation):

$$I(\Omega, Z_t \mid S_t, S_0 = s) \leq \mathbb{E}_{\Omega \sim p(\omega), S_t \sim p^J(s_t \mid \Omega, S_0 = s), Z_t \sim p(z_t \mid S_t, \Omega)} \left[ \log \frac{p(Z_t \mid \Omega, S_t)}{q(Z_t)} \right], \quad (5)$$

where $q(z_t)$ is a fixed variational approximation (set to $\mathcal{N}(0, \mathbf{I})$ as in InfoBot), and $p_\phi(z_t \mid \omega, s_t)$ is a parameterized encoder. As explained in section 1 the key difference between eq. (5) and InfoBot is that they construct upper bounds on $I(G, A_t \mid S_t, S_0)$, while we bottleneck the option-information.

We can compute a Monte Carlo estimate of Equation 5 by first sampling an option $\omega$ at $s_0$ and then keeping track of all states visited in trajectory $\tau$. In addition to the VIC term and our bottleneck regularizer, we also include the entropy of the policy over the actions (maximum-entropy RL (Ziebart, 2010)) as a bonus to encourage sufficient exploration. Overall, the DS-VIC objective is:

$$\max_{\theta, \phi, \nu} \tilde{J}(\theta, \phi, \nu) = \mathbb{E}_{\Omega \sim p(\omega), \tau \sim p^J(\cdot \mid \Omega, S_0), Z_t \sim p_\phi(z_t \mid S_t, \Omega)}$$

$$\left[ \log \frac{q_\nu(\Omega \mid S_f, S_0)}{p(\Omega)} - \sum_{t=0}^{f-1} \left( \beta \log \frac{p_\phi(Z_t \mid S_t, \Omega)}{q(Z_t)} + \alpha \log \pi_\theta(A_t \mid S_t, Z_t) \right) \right] \quad (6)$$

where $\theta, \phi$ and $\nu$ are the parameters of the policy, latent variable decoder and the option inference network respectively. Note that we abuse the notation slightly in Equation 6 by writing $\tau \sim p^J(\cdot \mid \omega, s_0)$ to denote a trajectory sampled by following $\pi(\cdot \mid \omega, s_0)$. This is a sequential process where $Z_t \sim p_\phi(z_t \mid s_t, \omega), A_t \sim \pi_\theta(a_t \mid s_t, z_t)$ and $S_{t+1} \sim p(s_{t+1} \mid s_t, a_t)$ (state transition function). The first term in the objective ensures reliable control in terms of visitation over a diverse set of final states in the environment while learning options; the second term ensures *minimality* in using the options sampled to make decisions (taking actions) and the third provides an incentive for *exploration*. See algorithm 1 in appendix for more details on the entire training pipeline.

**Handling Partial Observability.** We are generally interested in identifying decision states in partially observable environments (similar to InfoBot Goyal et al. (2019)). To formalise how we handle this, let $o_t$ be the (partial) observation made by the agent at time $t$. We set the state $s_t = g(o_0, \cdots, o_t)$ in all the equations above, where $g$ is a recurrent neural network (RNN) (Hochreiter & Schmidhuber, 1997) encoding partial observations. We found it important to parameterize the policy (Figure 2, right) as: $\pi(a_t \mid \omega, s_t) = \int dz\, p(z \mid s_t, \omega) p(a_t \mid o_t, z)$, *i.e.* where action distribution $p(a_t \mid o_t, z)$ are modelled as 'reactive' (conditioned only on the current observation) but $z$ distributions $p(z_t \mid s_t, \omega)$ are 'recurrent' (conditioning on the entire history of observations via $g(\cdot)$). This is because the sequence of observations $\{o_1, \cdots, o_t\}$ might be informative of the option $\Omega$ being followed *i.e.* conditioning $p(a_t \mid \cdot)$ on $S_t$ in addition to $z$ could 'leak' information about the underlying option to the actions, making it the bottleneck penalty on $z$ potentially ineffective. Finally, when maximizing the likelihood of option $\Omega$ given the final state $S_f$ in learning $q(\omega \mid s_f)$ in VIC (the option inference network), we use the global (x, y) coordinates of the final and initial state of the agent. Note that the agent's policy continues to be based only on observations $o_t$, and it is only at the end of the trajectory that we use the global coordinates to compute intrinsic reward. Please refer to Sec. A.6 for a discussion of the issues when training the model with partial observations in the option inference network.

**Related Objectives.** Eysenbach et al. (2018) (DIAYN) attempts to learn skills (similar to options) which can control the states visited by agents while ensuring that states, not actions, are used to distinguish skills. Thus, for an option $\Omega$ and every state $S_t$ in a trajectory, they maximize $\sum_t I(\Omega, S_t) - I(A_t, \Omega \mid S_t) + H(A_t \mid S_t)$, as opposed to $I(\Omega, S_f) - \beta \sum_t I(A_t, \Omega \mid S_t) + H(A_t \mid S_t)$ in our objective. With the sum over all timesteps for $I(\Omega, S_t)$, the bound in theorem 2.1 no longer holds true, which also means that there is no principled reason (unlike our model) to scale the second term with $\beta$ – which we find to be crucial for identifying decision states (similar to findings in InfoBot). Experimentally, we show that the decision states with DIAYN are no longer as interpretable, since options in this case are more low level and correspond to trajectories instead of goals.

The most closely related work to ours is InfoBot, which maximizes $\sum_t R(t) - \beta I(A_t, G|S_t)$ for a goal ($G$) conditioned policy $\pi(a_t|S_t, G)$. The key difference is that InfoBot requires extrinsic rewards, while our work is strictly more general and scales even in the absence of extrinsic rewards.

Further, in context of both these works, our work provides a principled connection between action-option information regularization $I(A_t, \Omega|S_t)$ and empowerment of an agent. The tools from Theorem 2.1 might be useful for analysing these previous objectives which both employ this technique.

## 3 Experiments

### 3.1 Transfer to Goal-Driven Tasks

Do the unsupervised decision states meaningfully transfer to novel tasks? To address this, we replicate the InfoBot experimental setup, replacing decision state identification (pre-training) with DS-VIC.

**Exploration-based Transfer**. Goyal et al. (2019) pretrain their model to identify decision states, and then study if decision states improve exploration when training a new policy $\pi_\gamma(a|s, g)$ from scratch. Given an environment with reward $R_e(t)$, goal $G$, $\kappa > 0$, and state visitation count $c(S_t)$, their reward is:

$$R_t = R_e(t) + \frac{\kappa}{\sqrt{c(S_t)}} \underbrace{I(G, Z_t|S_t)}_{\text{Pretrained, Frozen}} \qquad (7)$$

The count-based reward[2] decays with square root of $c(S_t)$ to encourage the model to explore novel states, and the mutual information between goal $G$ and bottleneck variable $Z_t$ identifies decision states, and is multiplied with the exploration bonus to encourage visitation of decision states.

We use an almost identical setup, replacing their decision-state term from supervised pretraining with DS-VIC intrinsic reward pretraining:

$$R_t = R_e(t) + \frac{\kappa}{\sqrt{c(S_t)}} \underbrace{I(\Omega, Z_t|S_t, S_0)}_{\text{Pretrained, Frozen}} \qquad (8)$$

$I(\cdot)$ is computed with eq. (5) with a parameterized $p(z_t \mid \omega, s_t)$ encoder frozen during transfer. See Algorithm 1 in appendix for more details.

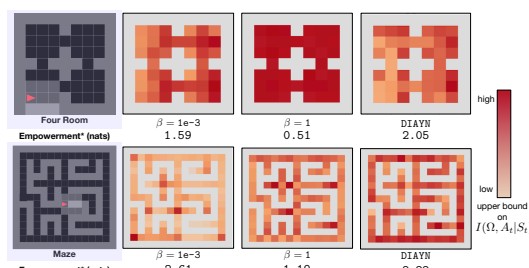

**Environments.** We pre-train and test on gridworlds from the MiniGrid (Chevalier-Boisvert et al., 2018) environments. We first consider a set of simple environments – `4-Room` and `Maze` (see Fig. 3) followed by the `MultiRoomNXSY` also used by Goyal et al. (2019). The `MultiRoomNXSY` environments consist of X rooms of size Y, con-

Figure 3: Decision state heatmaps plotting $I(\Omega, Z_t|S_t, S_0)$ at visited states on simple environments – 4-Room (top) and maze (bottom). First column depicts environment layout, second and third show results for DS-VIC for $\beta = 1e^{-3}$ and $\beta = 1$ respectively, and the fourth column shows DIAYN. Please refer to Appendix A.4 for details about how $I(\Omega, Z_t|S_t, S_0)$ is computed. * denotes lower bound (Eq. 1) on empowerment.

nected in random orientations across multiple runs. We also conduct experiments on `Mountain-Car` (Brockman et al., 2016), a continuous state-space environment with discrete actions. In all pre-training environments, options are executed for a fixed number of steps before terminating.

We use Advantage Actor-Critic (A2C) for all experiments. Since code for InfoBot (Goyal et al., 2019) was not public, we report numbers based on a re-implementation of InfoBot, ensuring consistency with their architectural and hyperparameter choices. We will release our code upon publication.

## 4 Results

**Baselines.** We evaluate the following on quality of exploration and transfer to downstream goal-driven tasks with sparse rewards: 1) InfoBot (our implementation) – which does goal-driven extraction of decision states, 2) DIAYN – whose focus is unsupervised skill acquisition (and not decision

---

[2]Visitation counts have several limitations including the requirement of a discrete state space and a state table for maintaining visit counts in POMDPs. However, our exploration bonus need not be restricted by these assumptions. One could alternatively use a technique similar to Burda et al. (2018) where the incentive is to learn to distill a trained encoder network, alleviating the need for visitation counts altogether.

states)[3], but has an $I(A_t, \Omega|S_t)$ term which can be used for the bonus in Equation 8, 3) count-based exploration which uses visitation counts as exploration incentive (this corresponds to replacing $I(\Omega, Z_t|S_t, S_0)$ with 1 in Equation 8), 4) a randomly initialized encoder $p(z_t \mid \omega, s_t)$ to check if learning in DS-VIC or InfoBot models improves over random initialization, 5) how different values of $\beta$ affect performance, and 6) a heuristic baseline that looks for local features like corners and doorways and uses every such occurrence as a decision state. This validates the extent to which relevant goal information is useful in identifying decision states, *vs.* local, feature-based heuristics.

## 4.1 QUALITATIVE RESULTS

*Grid Worlds:* Figure 3 shows results on 4-Room and maze. We notice that when $\beta = 1$, DS-VIC collapses to identifying every state as a decision state and gets poor empowerment: 0.51 (row 1, column 3 in Figure 3), indicating that the regularization strength is too high and hinders empowerment maximization. At lower values of $\beta = $ 1e-3, we get more discernible decision states. Across both environments, we notice that some corners and intersections are identified as decision states. In general, for DS-VIC on maze with $\beta = $ 1e-3 (when it gets an empowerment of 2.61), we notice that if an intersection is a decision state, and there is a corner next to it, it is often *not* a decision state. That is, not every state which looks like an corner needs to be a decision state, if for example the part of the state space in question is only spanned by one option. Finally, for maze we see that for a similar value of empowerment[4], DS-VIC leads to a sparser decision state distribution than DIAYN. This makes sense, since intuitively, one expects to have to make more decisions when following a particular trajectory (as in DIAYN), as opposed to a terminal state (as in DS-VIC).

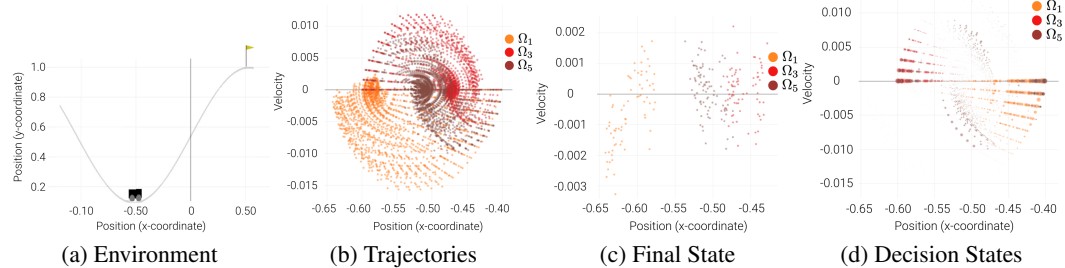

|       |       |       |       |
|-------|-------|-------|-------|
| (a) Environment | (b) Trajectories | (c) Final State | (d) Decision States |

Figure 4: We visualize identified decision states for 3 options by DS-VIC on the Mountain Car environment. (a) shows the environment layout, (b) the trajectories corresponding to the 3 options in position-velocity space – if trajectories corresponding to two different options reach the same position and velocity, then they are said to intersect, (c) the final-state distributions in the position-velocity space, and (d) the identified decision states. In (b), (c), (d), x-axis (position) shows the x-coordinate of the car, while y-axis shows the velocity of the car.

*Mountain Car Environment:* We also use DS-VIC to identify decision states in the Mountain Car (Brockman et al., 2016) environment – a continuous state-space (position and velocity of the car) environment with three discrete actions $\{+1(\texttt{right}), 0, -1(\texttt{left})\}$, which correspond to the direction of the force being applied to the car (see Fig. 4(a)). Fig. 4(b) shows the trajectories and Fig. 4(c) shows the final state(s) corresponding to 3 specific options (out of 8) overlaid on a position-velocity plane. Upon observing decision states in the same position-velocity plane (see Fig. 4(d), point-radius proportional to $I(\Omega, Z_t|s_t, s_0)$, we find that decision states occur at points in the state-space where trajectories associated with different options intersect. Intuitively, these are the states where the agent can switch from one option (mode of behavior) to another and therefore, needs to make a decision. Interestingly, there is a dominance of decision states along the $velocity = 0$ line, which indicates that the model identifies decision states to be points where the cart has $velocity = 0$ at some position (specific to the option). Finally, decision states often occur, for, say, the orange option ($\Omega_1$) in the right half of the position-velocity plane and never in the bottom-left quadrant, where it is the only option covering that part of the state-space, and thus default behavior can be practiced.

---

[3]Note that to keep things consistent with our method and InfoBot, we use $I(Z_t, \Omega|S_t)$ to impose the mutual information regularization in DIAYN, whereas Eysenbach et al. (2018) use a different construction.

[4]Since DIAYN maximizes the mutual information with every state in a trajectory, we report the empowerment for the state with maximum mutual information with the option.

| Method | MultiRoomN3S4 | MultiRoomN5S4 | MultiRoomN6S25 |
|---|---|---|---|
| (Encoder $p_\phi(Z_t|S_t, \Omega)$ pretrained on) | MultiRoomN2S6 | MultiRoomN2S6 | MultiRoomN2S10 |
| InfoBot (Goyal et al., 2019) | 90% | 85% | N/A |
| InfoBot (Our Implementation) | $99.6\%_{\pm 0.2\%}$ | $98.9\%_{\pm 0.8\%}$ | $86.8\%_{\pm 2.3\%}$ |
| Count-based Exploration (Our Implementation) | $99.8\%_{\pm 0.2\%}$ | $71.4\%_{\pm 28.1\%}$ | $88.4\%_{\pm 2.3\%}$ |
| DIAYN-based Exploration Bonus ($\beta = 1$) | $99.7\%_{\pm 0.2\%}$ | $\mathbf{99.7\%_{\pm 0.2\%}}$ | $0.1\%_{\pm 0.1\%}$ |
| Randomly Initialized Encoder | $100\%_{\pm 0\%}$ | $98.8\%_{\pm 0.7\%}$ | $67.9\%_{\pm 7.7\%}$ |
| Heuristic Decision States | N/A | N/A | $81.3\%_{\pm 3.0\%}$ |
| DS-VIC ($\beta = 10^{-3}$) | $99.9\%_{\pm 0.1\%}$ | $99.1\%_{\pm 0.3\%}$ | $90.9\%_{\pm 0.6\%}$ |
| DS-VIC ($\beta = 10^{-2}$) | $\mathbf{100\%_{\pm 0\%}}$ | $71.1\%_{\pm 28.1\%}$ | $\mathbf{92.6\%_{\pm 0.6\%}}$ |

Table 1: Success rate (mean $\pm$ standard error) of the goal-conditioned policy when trained with different exploration bonuses in addition to the extrinsic reward $R_e(t)$. We report results at $\sim 5 \times 10^5$ timesteps for MultiRoomN3S4, MultiRoomN5S4 and at $\sim 8.2 \times 10^6$ timesteps for MultiRoomN6S25. We also report the performance of InfoBot for completeness. Note that for rooms of size 4 (MultiRoomN3S4, MultiRoomN5S4), incentivizing to visit corners and doorways (Heuristic Decision States) is equivalent to count-based exploration.

## 4.2 QUANTITATIVE RESULTS

**Transfer to Goal-Driven Tasks**. Next, we evaluate Equation 8, *i.e.* whether incentivizing the agent to visit decision states in addition to extrinsic reward can aid in transfer to goal-driven tasks in different environments (see Sec. 3.1). We restrict ourselves to the point-navigation task (Goyal et al., 2019) transfer in the MultiRoomNXSY set of partially-observable environments. In this task, the agent is always spawned in the first room, and has to go to a randomly sampled goal location in the last room, and is rewarded only when it reaches the correct cell. Goyal et al. (2019) test the efficacy of different exploration objectives[5] and show that this is a hard setting where efficient exploration is necessary. They show that InfoBot outperforms several state-of-the art exploration methods in this environment.

Concretely, we 1) train DS-VIC to identify decision states (Equation 2) on MultiRoomN2S6 and transfer to a goal-driven task on MultiRoomN3S4 and MultiRoomN5S4 (similar to Goyal et al. (2019)), and 2) train on MultiRoomN2S10 and transfer to MultiRoomN6S25, which is a more challenging transfer task *i.e.* it has larger rooms (25x25) making efficient exploration critical to find doors quickly. For DS-VIC and DIAYN, we train for 6 million episodes in the decision state identification phase (pre-training) and pick the checkpoints with highest empowerment values across training.

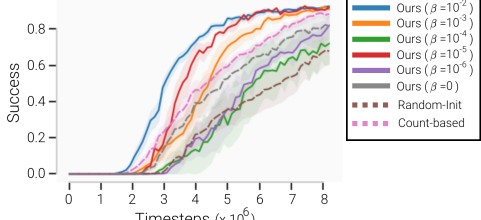

Figure 5: Effect of $\beta$ during DS-VIC pre-training on success with exploration bonus for the goal-conditioned policy. Shaded areas represent standard error of the mean over 10 random seeds.

**Overall Trends.** Table. 1 reports success rate – the % of times the agent reaches the goal on validation. First, our implementation of InfoBot is competitive with Goyal et al. (2019)[6]. For the MultiRoomN2S6 to N5S4 transfer (middle column), methods which identify decision states outperform count-based exploration. Interestingly, in this setup, a randomly initialized encoder is competitive with both InfoBot and DS-VIC, and serves as an important baseline. In MultiRoomN2S10 to N6S25 transfer, which has a large state space, we find that DS-VIC (at $\beta = 10^{-2}$) achieves the best performance, followed by count-based exploration and InfoBot (that are both within standard-error of each other). This shows that decision states identified by DS-VIC effectively guide exploration in sparse-reward settings. More importantly, our model beats a strong heuristic baseline which treats every corner and doorway as a decision state (more details on this are in the appendix). Finally, the randomly initialized encoder as well as DIAYN generalize much worse in this transfer task.

$\beta$ **sensitivity.** We sweep over $\beta$ in log-scale from $\{10^{-1}, \cdots, 10^{-6}\}$, as shown in Figure 5 (except $\beta = 10^{-1}$ which does not converge to $> 0$ empowerment) and also report $\beta = 0$ which recovers a no information regularization baseline. We find that $10^{-2}$ works best, followed by $10^{-5}$ and $10^{-3}$. In general, performance is robust, *i.e.* across $\beta$, we seem to reach $\approx 90\%$ success for most values.

---

[5] While our focus is on identifying and probing how good the decision states from intrinsic training are, more broader comparisons to exploration baselines are in InfoBot (Goyal et al., 2019) (reproduced in the appendix).

[6] We found it important to run all models (inlcuding InfoBot) an order of magnitude more steps compared to Goyal et al. (2019), but our models also appear to converge to higher success values.

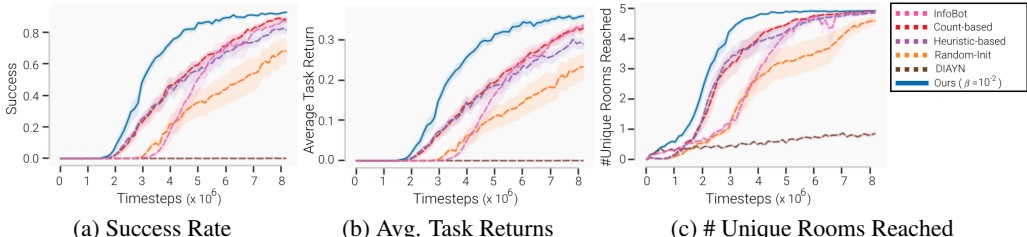

(a) Success Rate     (b) Avg. Task Returns     (c) # Unique Rooms Reached

Figure 6: Transfer results on MultiRoomN6S25 after unsupervised pre-training to identify decision states on MultiRoomN2S6. Shaded regions represent standard errors of the mean over 10 random seeds.

**More results on N6S25.** To better understand Equation 8 compared to other exploration incentives, in addition to success rate and sample-efficiency (Figure 6), we also consider another metric where we count the number of unique rooms reached (other than the starting room 0) as training progresses (Figure 6 (c)). Across metrics, DS-VIC performs the best on the challenging N6S25 transfer. Furthermore, the unique room metric reveals that DIAYN gets stuck after exploring the first room.

## 5 RELATED WORK

**Intrinsic Control and Intrinsic Motivation.** Learning how to explore without extrinsic rewards is a foundational problem in Reinforcement Learning (Machado et al., 2017; Pathak et al., 2017; Gregor et al., 2016; Strehl & Littman, 2008; Schmidhuber, 1990). Typical curiosity-driven approaches attempt to visit states that maximize the surprise of an agent (Pathak et al., 2017) or improvement in predictions from a dynamics model (Stadie et al., 2015; Lopes et al., 2012). While curiosity-driven approaches seek out and explore novel states, they typically do not measure how reliably the agent can reach them. In contrast, approaches for intrinsic control (Eysenbach et al., 2018; Achiam et al., 2018; Gregor et al., 2016) explore novel states while ensuring those states are reliably reachable. Gregor et al. (2016) maximize the number of final states that can be reliably reached by the policy, while Eysenbach et al. (2018) distinguish an option at every state along the trajectory, and Achiam et al. (2018) learn options for entire trajectories by encoding the sequence of states at regular intervals. Since decision states are more related to reliably acting in an environment rather than just visiting novel states (without an estimate of reachability), we formulate our regularizer in the intrinsic control framework, specifically building on the work of Gregor et al. (2016).

**Default Behavior and Decision States.** Recent work in policy compression has focused on learning a *default policy* when training on a family of tasks, to be able to re-use behavior across tasks. In (Galashov et al., 2018; Teh et al., 2017), default behavior is learnt using a set of task-specific policies which then regularizes each policy, while Goyal et al. (2019) learn a default policy using an information bottleneck on task information and a latent variable the policy conditions on. We devise a similar information regularization objective that learns default behavior shared by all intrinsic options without external rewards so as to reduce learning pressure on option-conditioned policies.

**Bottleneck states in MDPs.** There is rich literature on identification of bottleneck states in MDPs. The core idea is to either identify all the states that are common to multiple goals in an environment (McGovern & Barto, 2001) or use a diffusion model built using an MDP's transition matrix (Machado et al., 2017; Şimşek & Barto, 2004; Theocharous & Mahadevan, 2002). The key distinction between bottleneck states and decision states is that the latter are more closely tied to the information available to the agent and what it can act upon, whereas bottleneck states are more tied to the connectivity structure of an MDP and intrinsic to the environment, representing states which when visited allow access to a novel set of states (Goyal et al., 2019). Concretely, while a corner in a room need not be a bottleneck state, since visiting a corner does not 'open up' new states (the way a doorway would), it is still a useful state for a goal-driven agent with partial observation to visit (since it is a distinct landmark in the state space where meaningful decisions can be made).

**Information Bottleneck in Machine Learning.** Since the foundational work of Tishby et al. (1999); Chechik et al. (2005), there has been a lot of interest in making use of ideas from information bottleneck (IB) for various tasks such as clustering (Strouse & Schwab, 2017; Still et al., 2004), sparse coding (Chalk et al., 2016), classification using deep learning (Alemi et al., 2016; Achille & Soatto, 2016), cognitive science and language (Zaslavsky et al., 2018) and reinforcement learning (Goyal et al., 2019; Galashov et al., 2018; Strouse et al., 2018). We apply an information regularizer to an RL agent that must learn without extrinsic rewards to identify decision states.

# 6 CONCLUSION

We devise a principled approach to identify decision states in an environment without any extrinsic reward supervision using a sandwich bound on the empowerment of Gregor et al. (2016). Our approach yields decision states that align with human intuition across environments, and aid directed exploration on external-reward tasks and subsequently lead to better success rate and sample complexity in novel environments (competitive to supervised decision states of Goyal et al. (2019)). All our code and environments will be made publicly available.

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

# A    APPENDIX

## A.1    ALGORITHM

We present the algorithm summarizing our proposed approach DS-VIC– (1) unsupervised discovery of decision states and (2) transfer to goal-driven tasks.

---

**Algorithm 1** DS-VIC

---
**Require:**     A parameterized encoder $p_\phi(z_t \mid \omega, s_t)$, policy $\pi(a_t \mid \omega, s_t)$
**Require:**     A parameterized option inference network $q_\nu(\omega \mid s_0, s_f)$
**Require:**     A parameterized goal-conditioned policy $\pi_\gamma(a_t|s_t, g)$
**Require:**     A prior on discrete options $p(\omega)$ and integer $H$ - the length of each option trajectory.
**Require:**     A variational approximation of the option-marginalized encoder $q(z_t)$
**Require:**     A regularization weight $\beta$ and max-ent coefficient $\alpha$
**Require:**     A set of training environments $p_{\texttt{train}}(E)$ and transfer environments $p_{\texttt{transfer}}(E)$
   **Unsupervised Discovery**
   Sample training environment $E_{\texttt{train}} \sim p_{\texttt{train}}(E)$
   **for** episodes = 1 to $\texttt{max} - \texttt{episodes}$ **do**
      Sample a spawn location $S_0 \sim p(s_0|E_{\texttt{train}})$ and an option $\Omega \sim p(\omega)$
      Unroll a state-action trajectory $\tau$ under $\pi_\theta(a_t|s_t, z_t)$ for $H$ steps with reparametrized $Z_t \sim p_\phi(z_t|s_t, \omega)$
      Infer $\Omega$ from $q_\nu(\omega|s_o, s_f)$
      Update the parameters $\theta, \nu$ and $\phi$ based on Eqn. 6
   **end for**
   **Transfer to Goal-Driven Tasks**
   Sample transfer environment $E_{\texttt{transfer}} \sim p_{\texttt{transfer}}(E)$
   **for** episodes = 1 to $\texttt{max} - \texttt{episodes}$ **do**
      Sample a goal $G \sim p(g|E_{\texttt{transfer}})$
      Unroll a state-action trajectory under the goal-conditioned policy $\pi_\gamma(a_t|s_t, g)$
      Update policy parameters $\gamma$ to maximize the reward given by Eqn. 8
   **end for**

---

## A.2    LEMMA PROOF

We state a proof of the Lemma 2.1 stating that our proposed regularizer $\sum_{t=1}^{f} I(\Omega, A_t|S_t, S_0)$ forms an bound on empowerment $I(\Omega, S_f|S_0)$ from the main paper. This, combined with the lower-bound presented in VIC (Gregor et al., 2016), forms a sandwich bound on $I(\Omega, S_f|S_0)$.

**Lemma 2.1** Let $A_t$ be the action random variable at timestep $t$ and state $S_t$ following an option-conditioned policy $\pi(a_t|s_t, \omega)$. Then, $I(\Omega, A_t|S_t, S_0)$ *i.e.* the conditional mutual information between the option $\Omega$ and action $A_t$ when summed over all timesteps in the trajectory, upper bounds the conditional mutual information $I(\Omega, S_f|S_0)$ between $\Omega$ and the final state $S_f$ – namely the empowerment as defined by Gregor et al. (2016):

$$I(\Omega, S_f|S_0) \leq \sum_{t=1}^{f} I(\Omega, A_t|S_t, S_0) = \mathcal{U}_{DS}(\boldsymbol{\tau}, \Omega, S_0) \tag{9}$$

**Proof.** To begin, observe that the graphical model presented in Fig. 2 satisfies the markov chain $\Omega \leftrightarrow \{s_1, a_1\}_{t=1}^{f-1} \leftrightarrow S_f$ (assuming every node is conditioned on the intitial state $S_0$). Therefore, the data-processing inequality (DPI) Cover & Thomas (1991) implies:

$$I(\Omega, S_f|S_0) \leq I(\Omega, (S_{f-1}, A_{f-1})|S_0) \tag{10}$$

Furthermore, using the chain rule of mutual information Cover & Thomas (1991), we can write:

$$I(\Omega, S_f|S_0) \leq I(\Omega, (S_{f-1}, A_{f-1})|S_0) = I(\Omega, S_{f-1}|S_0) + I(\Omega, A_{f-1}|S_{f-1}, S_0) \tag{11}$$

Repeating the same set of steps recursively gives us:

$$
\begin{aligned}
I(\Omega, S_f|S_0) &\leq I(\Omega, (S_{f-1}, A_{f-1})|S_0) && \text{DPI (Cover \& Thomas, 1991)} && (12a) \\
&= I(\Omega, S_{f-1}|S_0) + I(\Omega, A_{f-1}|S_{f-1}, S_0) && \text{Chain rule of MI} && (12b) \\
&= I(\Omega, S_{f-2}|S_0) + I(\Omega, A_{f-2}|S_{f-2}, S_0) && && \\
&\qquad + I(\Omega, A_{f-1}|S_{f-1}, S_0) && \text{Same as 12a-12b} && (12c) \\
&\cdots && && (12d) \\
&= I(\Omega, S_0|S_0) + \sum_{t=1}^{f} I(\Omega, A_t|S_t, S_0) && && (12e) \\
&&&&& (12f)
\end{aligned}
$$

Note that the graphical model presented in Fig. 2 implies that $\Omega \perp\!\!\!\perp S_0$ and hence,

$$
I(\Omega, S_0|S_0) = H(\Omega) - H(\Omega|S_0) = H(\Omega) - H(\Omega) = 0 \tag{13}
$$

$$
\Rightarrow I(\Omega, S_f|S_0) \leq \sum_{t=1}^{f} I(\Omega, A_t|S_t, S_0) \tag{14}
$$

### A.3 Upper bound on $I(A_t, \Omega|S_t, S_0)$

We explain the steps to derive Eqn. (4) in the main paper, as an upper bound on $I(A_t, \Omega|S_t, S_0)$. By the data processing inequality (Cover & Thomas, 1991) $I(A_t, \Omega|S_t, S_0) \leq I(Z_t, \Omega|S_t, S_0)$ for the graphical model in Fig 2. So we will next derive an upper bound on $I(Z_t, \Omega|S_t, S_0)$.

We write $I(\Omega, Z_t|S_t, S_0 = s)$, given a start state $S_0 = s$ as:

$$
\mathbb{E}_{\Omega \sim p(\omega), S_t \sim p^J(s_t|\Omega, S_0=s), Z_t \sim p(z_t|S_t, \Omega)} \left[ \log \frac{p(Z_t|\Omega, S_t)}{p(Z_t|S_t)} \right] \tag{15}
$$

The key difference here is that our objective here uses the options $\Omega$ *internal* to the agent as opposed to Goyal et al. (2019), who use external goal specifications $g$ provided to the agent. Similar to VIC, $p^J$ here denotes the (unkown) state distribution at time $t$ from which we can draw samples when we execute a policy.

We then assume a variational approximation $q(z_t)$[7] for $p(z_t|S_t)$, and using the fact that $D_{\text{KL}}[p(z_t|s_t)||q(z_t)] \geq 0$, we get a the following lower bound:

$$
I(\Omega, Z_t|S_t, S_0 = s) \geq \mathbb{E}_{\Omega \sim p(\omega), S_t \sim p^J(s_t|\Omega, S_0=s), Z_t \sim p(z_t|S_t, \Omega)} \left[ \log \frac{p(Z_t|\Omega, S_t)}{q(Z_t)} \right] \tag{16}
$$

### A.4 Decision State Discovery – On policy with Options.

We use Eqn. 16 to discover and visualize the decision states learned in an environment, augmented with random sampling of the start state $S_0$. Thus, we compute our decision states in an on-policy manner. Mathematically, we can write this as:

$$
\mathbb{E}_{\Omega \sim p(\omega), S_0 \sim p(s_0), S_t \sim p^J(s_t|\Omega, S_0), Z_t \sim p(z_t|S_t, \Omega)} \left[ \log \frac{p(Z_t|\Omega, S_t)}{q(Z_t)} \right] \tag{17}
$$

where $S_0$ is a random spawn location uniformly chosen from the set of states in the environment and $\Omega$ is a random option chosen at each of the spawn locations. Thus, for each state $S_t$ in the environment, we look at the aggregate of all the trajectories that pass through it and compute the values of $\log \frac{p(z_t|\omega, s_t)}{q(z_t)}$ to identify / visualize decision states. In addition to being principled, this methodology also precisely captures our intuition that it is possible to identify decision states which are common to, or frequented across multiple options. Our results section 4 show that the identified decision states match our assessments of decision states corresponding to some structural regularities in the environment.

---

[7]For our experiments, we fix $q(z_t)$ to be a unit gaussian, however it could also be learned.

## A.5 IDENTIFICATION OF DECISION STATES DURING TRANSFER

As mentioned in the main paper, we would like to compute $I(\Omega, Z_t|S_t, S_0)$ to identify decision states and given a (potentially) novel environment and a novel goal-conditioned task, to provide it an exploration bonus. Given a state $s'$, that we would like to compute $I(\Omega, Z_t|S_t = s', S_0)$, we can write:

$$\sum_{\omega, s_0} \int dz_t \, p(\omega) p(s_0) p^J(S_t = s'|s_0, \omega) p(z_t|s_t, \omega) \log \frac{p(z_t|\omega, S_t = s')}{q(z_t)} \qquad (18)$$

However, this cannot be computed in this form in a transfer task, since a goal driven agent is not following on-policy actions for an option $\Omega$ that would allow us to draw samples from $p^J(\cdot|S_0, \Omega)$ above (in order to do a monte-carlo estimate of the integral above). Thus instead, we propose to compute the mutual information as follows:

$$\sum_{\omega, s_0} \int dz_t \, p(s_0, \omega|S_t = s') p(z_t|S_t = s', \omega) \log \frac{p(z_t|\omega, S_t = s')}{q(z_t)} \qquad (19)$$

Now, given a state $S_t = s'$, this requires us to draw samples from $p(s_0, \omega|S_t = s')$, which in general is intractable (since this requires us to know $p^J(s_t|\Omega)$, which is not available in closed form). In order to compute the above equation, we make the assumption that $p(s_0, \omega|S_t = s') = p(s_0)p(\omega)$. Breaking it down, this makes two assumptions: firstly, $p(\omega|S_t = s', s_0) = p(\omega|s_0)$. This means that all options have equal probability of passing through a state $s'$ at time $t$, which is not true in general. The second assumption is the same as VIC, namely that $p(\omega|s_0) = p(\omega)$. Instead of making this assumption, one could also fit a parameterized variational approximation to $p(\omega|S_t = s', s_0)$ and train it in conjunction with VIC. However, we found our simple scheme to work well in practice, and thus avoid fitting an additional variational approximation.

## A.6 DECISION STATES FROM LOCAL OBSERVATIONS

In the main paper, our results assumed that one had access to a good SLAM routine that did mapping and localization from partial observations $\{u_0, \cdots, u_t\}$. In general, as we discuss in the main paper, this is not a bad assumption since our model is always trained on a single environment, and thus it is reasonable for us to expect it to have a sense of the (x, y) coordinates in its internal representation / weights as training progresses. In this section, we detail some of the pathologies that we observed when kicking off training, if we took a naive approach to SLAM, estimating a function $f(\{u_0, \cdots, u_t\})$ to directly regress to $\Omega$ from partial observations (Algorithm 1). We note that since our particular choice of environment is a discrete gridworld, several partial observations (especially for small agent view sizes like the 3x3 window) look the same to the agent and the agent has a tendency to converge to the most trivial optima for learning options – one which learns to end at an easily obtainable partial observation given an option. For instance, while training DS-VIC, one particular policy simply learned 4 options corresponding to the four cardinal directions the agent can face, and each can be achieved by left or right turns, without actually requiring any movement by the agent. Since the agent has a compass which tells it the direction it is facing, it can ignore the partial observation and just use the direction vector to predict which of the 4 options was sampled.

## A.7 BASELINES

We use an exploration bonus coefficient of $\kappa = 0.1$ for the count-based exploration bonus baseline. The heuristic baseline identifies all occurrences of the following types of states in the MultiRoom environments – (1) corners of the room, (2) doorways, and gives them a slightly higher coefficient of exploration bonus than the $\kappa$ used for the count-based bonus. We ran a sweep for the values of this higher coefficient and found that $0.105$ (i.e. a $+5\%$ increase) gave best results.

## A.8 IMPLEMENTATION DETAILS

**Network Architecture:** We use a 3 layered convolutional neural network with kernels of size 3x3, 2x2 and 2x2 in the 3 layers respectively to process the agent's egocentric observation. We use ReLU as the non-linear activation function after each convolutional layer. The output of the CNN is then concatenated with the agent's direction vector (compass) and the option (or goal encoding). The concatenated features are then passed through a linear layer with hidden size 64 to produce

the final features used by option encoder $p_\phi(z_t|s_t, \omega)$ head and the policy head $\pi_\theta(a_t|s_t, z_t)$. We use the (x, y) coordinates of the final state (embedded through a linear layer of hidden size 64) to regress to the option via $q_\nu(\omega|s_0, s_f)$. Furthermore, our parameterized policy is a reactive one and the encoder $p_\phi(z_t|s_t, \omega)$ is recurrent over the sequence of states encountered in the episode. The bottleneck random variable $Z_t$ is sampled from the parameterized gaussian $p_\phi(z_t|s_t, \omega)$ and is made a differentiable stochastic node using the re-parmaterization trick for gaussian random variables.

**Training Details:** We use Advantage Actor-Critic (A2C) (open-sourced implementation by Kostrikov (2018)) for all our experiments. We use RMSprop as the optimizer for all our experiments. For the partially-observable grid world settings, the agent receives an egocentric view of it's surroundings as input, encoded as an occupancy grid where the channel dimension specifies whether the agent or an obstacle is present at an $(x, y)$ location. We set the coefficient $\alpha$ in Eqn. 6 to be $10^{-3}$ for all our experiments based on sweeps conducted across multiple values. In practice, we found it to be difficult to optimize our unsupervised objective in absence of an entropy bonus – the parameterized policy collapses to a deterministic one and no options are learned in addition to inefficient exploration of the state-space. Moreover, we found that it was hard to obtain reasonable results by optimizing for both the terms in the objective from scratch and therefore, we optimize the intrinsic objective itself for $\sim 8k$ episodes (i.e., we set $\beta = 0$) after which we turn on the regularization term and let $\beta$ grow linearly for another $\sim 8k$ episodes to get feasible outcomes at convergence. We experiment with a vocabulary of 2, 4 and 32 options (imitating undercomplete and overcomplete settings) for all the environments. For all the exploration incentives presented in Table 1, we first picked a value of $\kappa$ in Eqn. 7 (decides how to weigh the bonus with respect to the reward) from $\{10, 1, 0.1, 0.01, 0.001, 0.0001\}$ based on the best sample-complexity on goal-driven transfer.

**InfoBot Implementation:** Since code to reproduce InfoBot (Goyal et al., 2019) was not publiclt available, we implemented InfoBot ourselves while making sure that we are consistent with the architectural and hyper-parameter choices adopted by InfoBot (as per the A2C implementation by (Chevalier-Boisvert et al., 2018)). (i) We replace the convolution layers with fully connected layers to process the observations, and (ii) we use the layer sizes as mentioned by (Goyal et al., 2019) in the appendix. The checkpoint used to report the final performance for N3S4 and N5S4 were picked by doing validation on the transfer task based on the success metric. However, this procedure was infeasible for the bigger N6S25 environment and we chose the last checkpoint from the training for this environment.

**Convergence Criterion.** In practice, we observe that it is hard to learn a large number of discrete options using the unsupervised objective. From the entire option vocabulary, the agent only learns a few number of options discriminatively (as identified by the option termination state), with the rest collapsing into the same set of final states. To pick a suitable checkpoint for the transfer experiments, we pick the ones which have learned the maximum number of options reliably – as measured by the likelihood of the correct option from the final state $\log q_\nu(\omega|s_0, s_f)$.

**Option Curriculum** It is standard to have a fixed-sized discrete option space $\Omega$ with a uniform pior (Gregor et al., 2016). However, learning a meaningful option space with larger option vocabulary size $|\Omega| = K$ has been reported to be difficult Achiam et al. (2018). We adopt a curriculum based approach proposed in Achiam et al. (2018) where the option vocabulary size is gradually increased as the option decoder $q_\nu(\omega|s_f)$ becomes more confident in mapping back the final state back to the corresponding option sampled at the beginning of the episode. More concretely, whenever $q_\nu(\omega|s_0, s_f) > 0.75$ (threshold chosen via hyperparameter tuning), the option vocabulary size increases according to

$$K \leftarrow \min\left( \text{int}\left( 1.5 \times K + 1\right), K_{max}\right)$$

For our experiments, we start from $K = 2$ and terminate the curriculum when $K = K_{max} = 32$.

We run each of our experiments on a single GTX Titan-X GPU, and use no more than 1 GB of GPU memory per experimental setting.

## A.9 ADDITIONAL RESULTS

In Fig. 7 and Fig. 8, we show further results on the MultiRoomN3S4 and MultiRoomN5S4 environments to compare DS-VIC with other exploration incentives in terms of sample-complexity and the quality of the exploration incentive. As seen in the figures, DS-VIC is either competitive with or better than the presented baselines. As highlighted before, MultiRoomN3S4 and MultiRoomN5S4

| Method | MultiRoomN3S4 | MultiRoomN5S4 | MultiRoomN6S25 |
|---|---|---|---|
| (Encoder $p_\phi(Z_t|S_t, \Omega)$ pretrained on) | MultiRoomN2S6 | MultiRoomN2S6 | MultiRoomN2S10 |
| Goal-conditioned A2C | $0\%$ | $0\%$ | $0\%$ |
| TRPO + VIME | $54\%$ | $0\%$ | N/A |
| Curiosity-based Exploration | $95\%$ | $54\%$ | N/A |
| Count-based Exploration Goyal et al. (2019) | $95\%$ | $0\%$ | N/A |
| InfoBot Goyal et al. (2019) | $90\%$ | $85\%$ | N/A |
| InfoBot (Our Implementation) | $99.6\%_{\pm0.2\%}$ | $98.9\%_{\pm0.8\%}$ | $86.8\%_{\pm2.3\%}$ |
| Count-based Exploration (Our Implementation) | $99.8\%_{\pm0.2\%}$ | $71.4\%_{\pm28.1\%}$ | $88.4\%_{\pm2.3\%}$ |
| DIAYN-based Exploration Bonus ($\beta = 1$) | $99.7\%_{\pm0.2\%}$ | $\mathbf{99.7\%}_{\pm0.2\%}$ | $0.1\%_{\pm0.1\%}$ |
| Randomly Initialized Encoder | $100\%_{\pm0\%}$ | $98.8\%_{\pm0.7\%}$ | $67.9\%_{\pm7.7\%}$ |
| Heuristic Decision States | N/A | N/A | $81.3\%_{\pm3.0\%}$ |
| DS-VIC ($\beta = 10^{-3}$) | $99.9\%_{\pm0.1\%}$ | $99.1\%_{\pm0.3\%}$ | $90.9\%_{\pm0.6\%}$ |
| DS-VIC ($\beta = 10^{-2}$) | $\mathbf{100\%}_{\pm0\%}$ | $71.1\%_{\pm28.1\%}$ | $\mathbf{92.6\%}_{\pm0.6\%}$ |

Table 2: Success rate (mean $\pm$ standard error) of the goal-conditioned policy when trained with different exploration bonuses in addition to the extrinsic reward $R_e(t)$. We report results at $\sim5 \times 10^5$ timesteps for MultiRoomN3S4, MultiRoomN5S4 and at $\sim8.2 \times 10^6$ timesteps for MultiRoomN6S25. We also report results from InfoBot (Goyal et al., 2019) for completeness. Note that for rooms of size 4 (MultiRoomN3S4, MultiRoomN5S4), incentivizing to visit corners and doorways (Heuristic Decision States) is equivalent to count-based exploration.

are relatively simpler environments where simpler baselines like count based exploration incentives also perform quite well due to the small size of rooms.

In Table. 2, we report broader comparisons with other exploration baselines InfoBot (Goyal et al., 2019) compare with.

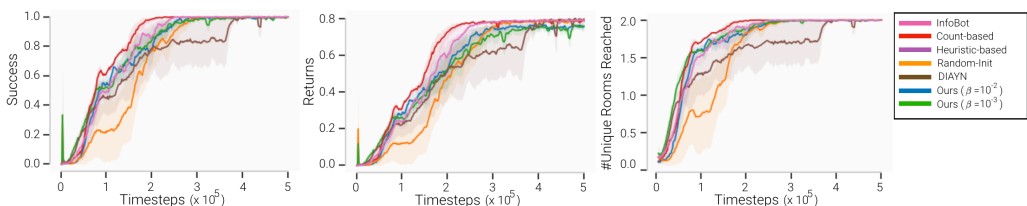

(a) Success Rate   (b) Avg. Task Returns   (c) Farthest Room Reached (running best)

Figure 7: (a) Success Rate, (b) Avg. Task Returns, (c) farthest room reached so far, versus time-steps on MultiRoomN3S4. Unsupervised pre-training to identify decision states was done on MultiRoomN2S6. Shaded areas represent standard error of the mean over 3 random seeds.

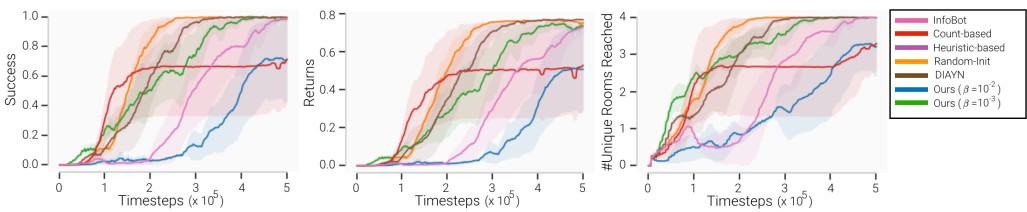

(a) Success Rate   (b) Avg. Task Returns   (c) Farthest Room Reached (running best)

Figure 8: (a) Success Rate, (b) Avg. Task Returns, (c) farthest room reached so far, versus time-steps on MultiRoomN5S4. Unsupervised pre-training to identify decision states was done on MultiRoomN2S6. Shaded areas represent standard error of the mean over 3 random seeds.

