# OpenReview forum: "DS-VIC: Unsupervised Discovery of Decision States for Transfer in RL"
_ICLR.cc/2020/Conference — Reject_

### Official Review · AnonReviewer1 · 2019-10-22
**Official Blind Review #1**

**Rating:** 3

**Review:**

Summary:

This paper proposes an unsupervised method for discovering "decision states", defined as states where decisions affect the future states an agent can reach in the environment, based on the variational intrinsic control (VIC) framework that maximizes an agent's empowerment. This paper draws connections to many prior works such as VIC, diversity is all you need (DIAYN), and InfoBot and shows results on MiniGrid and MountainCar.

Main Comments:

While this is an interesting paper, I did not find the experimental section to be convincing enough for publication at this stage. Moreover, I am concerned by the novelty of the proposed approach, which seems very similar to InfoBot, the main difference between them being the replacement of the goals with options thus moving towards less supervision / use of prior-knowledge. However, if I understand correctly, this method still requires to specify a prior over the options, so it is not clear why DS-VIC would be preferable to InfoBot. If the empirical results showed a more robust and significant gain in performance on more diverse or complex tasks, I would be willing to reconsider my judgement regarding the significance of this work.

Minor Questions / Comments:

1. How do you define the final state S_f? Do you only consider the episodic RL setting? Do you consider S_f to always be after a fixed number of steps or whenever the termination function is triggered?

2. Please include more information about what is represented in Figure 3 and the color scale. I am slightly confused by the interpretation of that plot because (1) it seems like the model does not detect "all decision states" (e.g. intersections) . that a human may consider while including others (e.g. corners, for which I do not agree that the agent should be incentivized to go even after learning from the reward structure that there isn't much to gain), (2) why is it that the for example the top-left figure has a rather nonuniform distribution across the rooms (is it influenced by the initial position of the agent?) and (3) the model doesn't seem to be very consistent about what it considers a to be a "decision state".

3. Can you show similar plots for the MultiRoom environment? Perhaps those will shed more light into the learned behavior.

4. Including all possible ablations to the objective in equation 6 would be helpful to tease apart the contribution of each term: variational control, information bottleneck, and entropy.

5.  The results in Table 1 and Figures 6 do not show a significant gain in performance. Moreover, I suspect the other methods will converge to similar values soon after 8M steps. For a fair comparison, it would be good to show how the curves look after all (or at least . more of the models) converge. From Table 1, it actually seems to me that InfoBot encounters less penalty across all 3 models, even tho DS-VIC overperforms on the more challenging one. Moreover, in all 3 cases, at least one of the other methods seems to at least be close to the performance of DS-VIC so I am concerned that these may not be very challenging tasks for well-tuned SOTA methods. Plus, the comparison does not seem fair given that the the numbers are reported before the baselines converged.

6. Did you pretrain the baselines for the same number of steps as DS-VIC? Please include more details about this stage and how you ensure the comparison was fair.

7. It might be useful to include other baselines such as the curiosity-based exploration method from Pathak et al, 2017 . or universal value functions (Schaul et al. 2015)


**Experience Assessment:**

I have read many papers in this area.

**Review Assessment: Checking Correctness Of Derivations And Theory:**

I assessed the sensibility of the derivations and theory.

**Review Assessment: Checking Correctness Of Experiments:**

I assessed the sensibility of the experiments.

**Review Assessment: Thoroughness In Paper Reading:**

I read the paper at least twice and used my best judgement in assessing the paper.

---

> ### Author Response · Authors · 2019-11-15
> **Response to AnonReviewer1  (Part 1/2)**
>
> > While this is an interesting paper, I did not find the experimental section to be convincing enough for publication at this stage. Moreover, I am concerned by the novelty of the proposed approach, which seems very similar to InfoBot, the main difference between them being the replacement of the goals with options thus moving towards less supervision / use of prior-knowledge. However, if I understand correctly, this method still requires to specify a prior over the options, so it is not clear why DS-VIC would be preferable to InfoBot
>
> Similar to modern approaches in deep variational inference (such as a variational auto-encoder [1]), the "prior" is something that is parameterized as a distribution in some abstract latent space and does not necessarily imply hardcoding some knowledge. In such models the prior generally gets "meaning" because of what the decoder (the policy in our case) ends up learning. We follow the same parameterization as DIAYN, setting the prior as a uniform categorical distribution. We have added this to the paper.
>
> As mentioned in introduction of the submission, DS-VIC is preferable to InfoBot in cases where specifying goals is difficult. For e.g. when: 1) rewards are sparse or absent (Pathak et al., 2017), 2) for an agent to learn meaningful behavior, proxy goals and rewards need to be hand engineered (making it hard to scale), and 3) the notion of a goal might not even be obvious in some cases.
>
> > If the empirical results showed a more robust and significant gain in performance on more diverse or complex tasks, I would be willing to reconsider my judgement regarding the significance of this work.
>
> We stand by our initial response to all reviewers that given the outlook of our paper to demonstrate utility of our notion of decision states as the marginal over options, we compared directly to the experimental setup of InfoBot. The tasks considered in the paper are goal-driven tasks with really sparse reward and are fairly complex in that sense, requiring appropriate exploration (Section 4.2 Transfer to Goal-Driven Tasks).
>
> > 1. How do you define the final state $S_f$? Do you only consider the episodic RL setting? Do you consider $S_f$ to always be after a fixed number of steps or whenever the termination function is triggered?
>
> Yes, we consider the episodic RL setting where each option is terminated after a fixed number of steps, which is a hyperparameter. We have added these details in the experiments section.
>
> > 2. Please include more information about what is represented in Figure 3 and the color scale.
>
> We have added the missing color scale in the revision. The darker shades of red represent higher values of $I(A_t, \Omega | S_t)$ and are supposed to represent decision states.
>
> > ... (1) it seems like the model does not detect "all decision states" (e.g. intersections) . that a human may consider while including others (e.g. corners, for which I do not agree that the agent should be incentivized to go even after learning from the reward structure that there isn't much to gain), ...
> > ... and (3) the model doesn't seem to be very consistent about what it considers a to be a "decision state".
>
> Please refer to our general response about alignment of decision states with human intuition. Moreover, subject to optimization and initialization the model identifies different options which reach different parts of the state space. The decision states are a function of the options learnt (in the manner we define them). Thus, each intersection need not be a decision state (if there is only one option that leads to that part of the space).
> In general, we do not claim that our method necessarily identifies all states that humans would agree as decision states but we find that the decisions state that do emerge have some non-trivial alignment with what humans would expect. Regardless, we show empirically that identifying decision states via the pre-trained encoder leads to better transfer performance in novel environments.
>
>
> > ... (2) why is it that the for example the top-left figure has a rather nonuniform distribution across the rooms (is it influenced by the initial position of the agent?) ...
>
> Yes, the decision states are influenced by the initial state of the agent and for the Four Room and Maze environments in Figure 3, the initial state was chosen uniformly at random from the set of all states.
>
> References
> [1]: Kingma, Diederik P., and Max Welling. 2013. “Auto-Encoding Variational Bayes.” http://arxiv.org/abs/1312.6114v10.

---

> > ### Author Response · Authors · 2019-11-15
> > **Response to AnonReviewer1  (Part 2/2)**
> >
> > > 3. Can you show similar plots for the MultiRoom environment? Perhaps those will shed more light into the learned behavior.
> >
> > Unfortunately, we did not find the decision states in this case to be human interpretable. However, these states are still "useful" as they lead to better transfer (which is ultimately our goal).
> >
> > > Including all possible ablations to the objective in equation 6 would be helpful to tease apart the contribution of each term: variational control, information bottleneck, and entropy.
> >
> > Please see our general response about the VIC + Maximum Entropy ablation. We do not drop the entropy term in our objective as we found it necessary for stable training of all option-based models.
> >
> > > 5. The results in Table 1 and Figures 6 do not show a significant gain in performance. Moreover, I suspect the other methods will converge to similar values soon after 8M steps. For a fair comparison, it would be good to show how the curves look after all (or at least . more of the models) converge. From Table 1, it actually seems to me that InfoBot encounters less penalty across all 3 models, even tho DS-VIC overperforms on the more challenging one... ... Plus, the comparison does not seem fair given that the the numbers are reported before the baselines converged.
> >
> > We trained all models for the same number of time-steps for a fair comparison and reported numbers after the best model converged. As a result, some of the baselines did not converge before the time-step limit. This demonstrated that DS-VIC had higher sample efficiency than baselines. Furthermore, in the MultiRoom N6S25 (6 rooms of max-size 25) environment a count-based baseline does converge asymptotically and no other baseline was able to beat its sample complexity except DS-VIC. While Table 1 evaluates all models trained for 8M steps (which may seem like an unfair comparison), Figure 6 demonstrates the sample efficiency of DS-VIC over baselines and justifies our choice of 8M steps.
> >
> > > ... Moreover, in all 3 cases, at least one of the other methods seems to at least be close to the performance of DS-VIC so I am concerned that these may not be very challenging tasks for well-tuned SOTA methods.
> >
> > We disagree, Fig. 6 clearly shows statistically significant improvements over baselines for DS-VIC.
> >
> > > 6. Did you pretrain the baselines for the same number of steps as DS-VIC? Please include more details about this stage and how you ensure the comparison was fair.
> >
> > Yes, we did train all baselines for equal number of steps but we picked best checkpoint over training for the transfer task (similar to Infobot). We have added these details in the revision.
> >
> > > It might be useful to include other baselines such as the curiosity-based exploration method from Pathak et al, 2017 . or universal value functions (Schaul et al. 2015)
> >
> > In general, curiosity based approaches are useful for high dimensional observation spaces such as images, and thus do not form a natural baseline for exploration in gridworlds. Moreover, InfoBot is a more natural baseline for our work than UVF. However, for completeness we plan to add both these baselines in the next revision.

---

### Official Review · AnonReviewer2 · 2019-10-23
**Official Blind Review #2**

**Rating:** 3

**Review:**

The authors introduce a novel decision point discovery method, wherein the VIC objective is constrained to minimize the amount of information between the option and the actions taken along the trajectory. After relaxing the constraint and introducing an upper bound to I(a; o), a tractable algorithm is produced. An implementation is then tested empirically on several partially observed grid worlds and a simple continuous control task on both qualitative bottleneck identification and quantitative benefits as an exploration bonus in a transfer learning setup.

Overall I think the approach is well motivated and interesting, but the resulting implementation takes too many unmotivated modifications to make work, and the results aren't terribly convincing despite this; as such I currently vote for it's rejection. Specifically, the usage of privileged information (x,y coordinates in what is described as a partially observed domain) and the ad hoc choice of which networks had memory (i.e. an LSTM) don't fit the narrative that motivates the work. Constraining the empowerment should be thing that handles spurious diversity, so the need to use x,y coordinates is concerning.

Regarding the empirical results, do all of the baseline make similar use of domain knowledge / privileged information? For example, does your implementation of DIAYN utilize x,y coordinates in the option predictor? Is the Beta=0 case considered? It isn't mentioned, but perhaps it amounts to one of your other baselines?

The empirical evidence isn't terribly convincing. On two of the three exploration setups, the random network is as performant, and does need a Beta hyper-parameter to tune. Though, to be fair, the connection between decision state identification and a good count-based exploration bonus is loose. The qualitative results are also a bit lacking. I was expecting the doorways to "pop out" more; the relatively muddled decision state activations made me wonder if they were really better than DIAYN's.

This work would really benefit from a quantitative measure of decision state identification accuracy. Some prior work (e.g. "Grounding Subgoals in Information Transitions") were able to do this by choosing environments where the quantities of interest were tractable to calculate exactly. This would at least allow us to see if the discovered decision states correspond to those that are optimal under your metric.

Rebuttal EDIT:
Thank you for the thoughtful rebuttal. If this were an option, I'd raise my score to a 5. But as my vote is to 'revise and resubmit' (unfortunately translated to 'reject' as per the conference system), I'll leave it in the 'reject' score bucket.

Your rebuttal lessened my concerns about the using (x,y) and only using an LSTM for the policy. I agree these are largely orthogonal issues, and since they were consistent with their baselines, that is fine.

However, the response to the unintuitive nature of the "decision states" is less convincing. If all you care about is the downstream task performance, why even show the qualitative results or impose the semantics of "decision states" on the learned representations? The sandwich bound is novel in and of itself; I understand the need to relate to prior work, but I actually think dropping the language around "decision states" (maybe outside of the algorithm's motivation) and talking purely in information theoretic terms would improve the paper.

Your response to [R3] on the setting of Beta hyper-parameter seems to not be supported by the results. You claim that values work well across multiple tasks, but the best reported value for the "hard" task (1e-2) is worse than the random baseline on the "easy" tasks.

Perhaps only the "hard" task matters and the "easy" tasks are only of significance due to their usage in InfoBot. But I'd argue that unless your method is dominating existing methods on both without changing hyper-parameters, switching to a more complex (and commonly used) benchmark would be more convincing.  The Atari Suite or the control tasks used in related work (e.g. DIAYN) would be my suggestion.

**Experience Assessment:**

I have published in this field for several years.

**Review Assessment: Checking Correctness Of Derivations And Theory:**

I assessed the sensibility of the derivations and theory.

**Review Assessment: Checking Correctness Of Experiments:**

I carefully checked the experiments.

**Review Assessment: Thoroughness In Paper Reading:**

I read the paper at least twice and used my best judgement in assessing the paper.

---

> ### Author Response · Authors · 2019-11-15
> **Response to AnonReviewer2**
>
> > the resulting implementation takes too many unmotivated modifications to make work, and the results aren't terribly convincing despite this ...
> > ... Specifically, the usage of privileged information (x,y coordinates in what is described as a partially observed domain) ...
>
> Constraining the empowerment should be thing that handles spurious diversity, so the need to use x,y coordinates is concerning.
>
> Please see our general response on the use of privileged (x, y) information.
>
> > Regarding the empirical results, do all of the baseline make similar use of domain knowledge / privileged information? For example, does your implementation of DIAYN utilize x,y coordinates in the option predictor?
>
> Yes, to ensure appropriate comparison, the relevant baselines also have access to privileged X-Y information during pre-training.
>
> > ... and the ad hoc choice of which networks had memory (i.e. an LSTM) don't fit the narrative that motivates the work
>
> The particular inductive biases on networks in deep learning (ex. convolution) generally have as big an impact as the exact objective and formulation that is used. We argue that our choice of an LSTM is also in the same spirit, and does not make the work any less principled. The difference between the recurrence used in InfoBot versus our model is that InfoBot uses recurrence over partially observed states for both the encoder $p(Z_t|S_t, G_t)$ and decoder $\pi(A_t|S_t, Z_t)$. Our model uses recurrence over just the encoder $p(Z_t|S_t, \Omega)$ and keeps the decoder \pi(A_t|S_t, Z_t)$ reactive. This choice was made because we do not have a time-varying goal vector that InfoBot had at every step, instead we have an episodic option which may be easily inferred from the past by the decoder.
>
> > Is the Beta=0 case considered?
>
> Please refer to the VIC + Max-Entropy baseline in our general response.
>
> > The empirical evidence isn't terribly convincing. On two of the three exploration setups, the random network is as performant, and does need a Beta hyper-parameter to tune
>
> The two smaller environments being pointed to — N5S4 and N3S4 — were used by InfoBot to report transfer performance. However, our point w.r.t. reporting the random network baseline was to show that these environments themselves are not complicated enough for any sophisticated approach to obtain significant (or meaningful) gains over a randomly initialized network. Therefore, in addition to these two, we also report performance on a larger environment — N6S25 — where there actually is some room for improvement beyond random baselines and heuristic based exploration strategies. Thus, we claim that our work actually establishes stronger baselines for the line of work that ours and InfoBot represents, and the random baseline is something that future works should also compare against to make meaningful progress.
>
> > Though, to be fair, the connection between decision state identification and a good count-based exploration bonus is loose
>
> As highlighted in Goyal et. al. (InfoBot), decision states represent a sparse set of sub-goals in the environment. A naive count-based exploration will encourage exhaustive visitation of the state space, whereas decision-states based exploration will narrow this to encourage visitation of a sparser set of decision states which would improve sample efficiency in hard exploration tasks. Informally, count-based methods perform “exhaustive exploration” whereas decision-state methods perform “targeted exploration” where the targets are all decision states.
>
> > The qualitative results are also a bit lacking. I was expecting the doorways to "pop out" more; the relatively muddled decision state activations made me wonder if they were really better than DIAYN's.
>
> Please refer to our general response about decision states aligning with human intuition.
>
> > This work would really benefit from a quantitative measure of decision state identification accuracy. Some prior work (e.g. "Grounding Subgoals in Information Transitions") ...
>
> We agree that Dijk & Polani [1] studied a simple 6-room environment where they were able to compute the Relevant Goal Information (RGI) explicitly using an optimal policy obtained by value iteration on the MDP. However, we note that all of our environments (including the Four Room and Maze in Fig. 3) are partially observed MDPs. If we were to compute the true quantities of interest assuming fully observed states, we would not expect alignment with what a partially observed agent ends up learning.
>
> References
> [1] van Dijk, Sander G., and Daniel Polani. "Grounding subgoals in information transitions." 2011 IEEE Symposium on Adaptive Dynamic Programming and Reinforcement Learning (ADPRL). IEEE, 2011.

---

### Official Review · AnonReviewer3 · 2019-10-23
**Official Blind Review #3**

**Rating:** 3

**Review:**

This paper proposes an algorithm for discovering decision states in MDPs, in a task agnostic manner. The proposed method essentially generalizes the information bottleneck approach used in InfoBot, to an unsupervised setting. Whereas InfoBot recovers decision states in goal-conditioned policies by minimizing mutual information between goals and actions, the proposed approach (DS-VIC) does the same using implicit goals discovered by variational intrinsic control. Concretely, the authors propose adding a regularization term which constrains the mutual information between an episodic latent variable (having high mutual information to a final state) and each action along the (option conditional) trajectory. This objective is shown to be equivalent to a constraint optimization problem, which fits both a lower and upper bound on the VIC mutual information term. On MiniGrid environments, the approach is shown to yield somewhat interpretable decision states. In the footsteps of InfoBot, the authors then show that the resulting regularizer (the “latent-action” mutual information term) can serve as a useful auxiliary reward for transfer tasks with a hard exploration problem (transfer from goal navigation in small to large rooms).

The paper is interesting and provides some interesting theoretical results which adds to the body of work on variational intrinsic control, and unsupervised reinforcement learning. To the best of my knowledge, the derived upper-bound to the mutual information term used in VIC is novel and would be of interest to the community. The extension of InfoBot to the unsupervised regime, swapping extrinsic goals for inferred options is also intuitive and generalizes published work.

That being said, I do not think the paper is ready for publication at this point in time.

On the experimental side, the results are limited and not entirely convincing. Experiments are unfortunately limited to MiniGrid environments and MountainCar, whereas most recent work on empowerment (DIAYN, VALOR, etc.) has focused on more complex continuous control benchmarks. In addition, the experiments themselves are limited in scope and rely mostly on evaluating whether the mutual information term between goal and actions can be used to craft an auxiliary reward for downstream tasks, a task first derived in the InfoBot paper. Unfortunately, the results here are mixed, with the mutual information based reward statistically outperforming count-based bonus (which it builds on) only when the optimal hyper-parameter $\beta$ (controlling the strength of the regularization term) is known. This somewhat breaks the narrative of unsupervised decision states. A much more compelling use case for the proposed regularization term would be improved data-efficiency on a downstreak task after pre-training with Eq. 6, following in the footsteps of DIAYN. Finally, as shown in the Appendix, the method seems rather brittle, requiring both a schedule on the size of the option layer and the strength of the regularization term, something which would be difficult to do in the absence of a downstream task.

There is also an important missing baseline which is glossed over. Instead of regularizing the mutual information between goals and actions $I(\Omega, A_t \mid S_t, S_0)$ one could simply encourage the low-level goal-conditioned policy to have high entropy. Indeed, one can show that  $KL[\pi(a_t \mid s_t, w_t) \| \pi_0(a_t)]$, with a learnt or fixed prior $\pi_0$, is an upper-bound to $I({s_t, w_t} ; a_t)$: hence minimizing this KL (equivalent to maximizing entropy) would naturally prevent high mutual information between options and individual actions. As this term is already present in Eq. 6 it would be interesting to repeat the experiments, dropping the second term (minimality) but instead sweeping over the strength of the entropy regularization term $\alpha$.

With respect to clarity, the paper could also be greatly improved. Decision states are never clearly defined to be those having high mutual information $I(\Omega, A_t \mid S_t)$. It is also not clear from the main text was is being plotted in Figure 3, requiring the reader to go through Appendix 4 to understand the visualization (without any references to this appendix in the main text). Similarly, I am not quite sure what to make of Figure 4 apart from the fact that different latent options yield different trajectories. See detailed comments below.


Detailed Comments:
(method)
* It is rather disappointing that the reverse predictor uses privileged information, in the guise of x-y features. This represents quite a lot of prior knowledge about what we wish the options to encode. How does the method perform from the raw state?
* Although mathematically elegant, I do not believe the sandwich bound explains why Eq. 6 helps uncover decision states. If we had an unbiased estimate of the mutual information between option and last state, then this would imply that an equality constraint on the VIC mutual information term would similarly yield decision states. This seems unlikely. An alternative hypothesis is that Eq. 6 works by injecting a soft prior, both via the temporal decomposition which aims to minimize $I(\Omega, A_t |...)$ and its upper-bound $I(\Omega, Z_t | …)$ which bottlenecks state information. Testing this theory could help strengthen the paper.
* It would be nice to spell-out that the standard “reverse bound” employed by VIC cannot be used to estimate $I(\omega, A_t \mid S_t, S_0)$ as this would yield a lower-bound whereas we aim to minimize this term. I was almost tricked into thinking this was a simpler and valid strategy before realizing my mistake.
* Footnote 7. High variance on $\beta=1e-4$. Is it possible that too few seeds were used to estimate the standard error on the mean?

(clarity)
*“Decision states to be points where the cart has velocity=0”: wouldn’t this mostly be restricted to the initial state?
What exactly was done for DIAYN in relation to Eq. 8? The text from S4-Baselines seems at odds with the caption.
* “Upper bound is too tight”. I don’t think this is what you mean: tight would refer to how good the upper-bound approximation is, which is different from the constraint specifying an upper-bound which is too small.
* Notation: Section 2.1 states that upper-case denotes random variables and lower-case denotes samples. Following this notation, equation should read e.g. $w \sim p(\Omega)$ and not $\Omega \sim p(w)$.
* Notation: $p^J(s_f \mid w, s_0)$. What does J refer to? J is not defined anywhere.
* Section 4.1: Did not understand the sentence “we noticed that if an intersection is a decision state [...] having already made the decision.”
* Section 4.1: Did not understand what is meant by “where trajectories associated with different options intersect”? What does this mean concretely in MountainCar for trajectories to intersect? Furthermore aren’t states with velocity=0 (mostly) restricted to the initial state?

**Experience Assessment:**

I have read many papers in this area.

**Review Assessment: Checking Correctness Of Derivations And Theory:**

I carefully checked the derivations and theory.

**Review Assessment: Checking Correctness Of Experiments:**

I carefully checked the experiments.

**Review Assessment: Thoroughness In Paper Reading:**

I read the paper thoroughly.

---

> ### Author Response · Authors · 2019-11-15
> **Response to AnonReviewer3 (Part 1/2)**
>
> We thank the reviewer for their detailed comments and feedback.
>
> > Experiments are unfortunately limited to MiniGrid environments and MountainCar, whereas most recent work on empowerment (DIAYN, VALOR, etc.) has focused on more complex continuous control benchmarks
>
> The focus in recent work (DIAYN, VALOR) on empowerment based objectives in RL is to identify / learn skills useful for downstream tasks. In contrast, as mentioned in the introduction, and in the general response, our focus is not on identifying transferable skills but rather using skills/options to identify "decision-states" in an environment (in an unsupervised manner) and study how well does providing an incentive to visit decision states aid transfer to novel environments.
>
> Moreover, the experiments in DIAYN and VALOR have an implicit assumption that skills have a common start state and due to their respective empowerment objectives, the skill-conditioned trajectories do not overlap after the first state. Hence, there is no obvious notion of what a “decision state” would look like in any of the DIAYN+VALOR experimental settings e.g.: Figure 2 (a, b) in the DIAYN paper has a single state common to all trajectories in the 2D navigation task and the first set of states in the overlapping skills task — where the skills do not overlap again after they leave the narrow region on the left.
>
> > the experiments themselves are limited in scope and rely mostly on evaluating whether the mutual information term between goal and actions can be used to craft an auxiliary reward for downstream tasks, a task first derived in the InfoBot paper
>
> We would like to reiterate that our goal was to demonstrate that a notion of decision states defined in terms of the marginal over different options can be useful for transfer to downstream tasks. Our mathematical formulation uses mutual information to formalize this notion of decision states, building on top of previous work such as [1].
>
> > the results here are mixed, with the mutual information based reward statistically outperforming count-based bonus (which it builds on) only when the optimal hyper-parameter $\beta$ (controlling the strength of the regularization term) is known
>
> The choice of $\beta$ is akin to model selection in any unsupervised learning algorithm e.g.: picking the number of clusters in K-means. We believe that using a particular supervised task to pick the value of \beta is not unfair, as long as the value of \beta thus selected transfers to other tasks. In response to this concern, we ran an experiment where we studied if the best value of \beta from N6S25 experiments also generalizes to other environments. Experimental evidence suggests this is true, with $\beta$ = 1e-2 yielding the transfer performance in MultiRoomN5S4 as well.
>
> > A much more compelling use case for the proposed regularization term would be improved data-efficiency on a downstreak task after pre-training with Eq. 6, following in the footsteps of DIAYN
>
> We ran such an experiment in order to inspect the performance of the learnt skills on continuous control tasks in the DIAYN setting with and without our information regularization. We ran it on Hopper, BipedalWalker, InvertedDoublePendulum and MountainCar (continuous actions) and found no statistically significant change in the performance and sample complexity of the best skill fine-tuned on external reward. Thus, it seems that the benefit from our approach is more in line with the gains from InfoBot, than those from DIAYN (in terms of transfer).
>
> > There is also an important missing baseline which is glossed over...
> > As this term is already present in Eq. 6 it would be interesting to repeat the experiments, dropping the second term (minimality) but instead sweeping over the strength of the entropy regularization term $\alpha$.
>
> Please refer to our general response, we have added this baseline with $\beta=0$. We found that sweeping $\alpha$ values generally does not lead to stable options learning for a broad range of alpha values. Thus, it is tougher to impose different strengths of the information regularization using this suggested parameterization and we instead fix a value of $\alpha$ and follow the InfoBot machinery to impose the bottleneck leveraging the DPI.
>
> [1] van Dijk, Sander G., and Daniel Polani. "Grounding subgoals in information transitions." 2011 IEEE Symposium on Adaptive Dynamic Programming and Reinforcement Learning (ADPRL). IEEE, 2011.

---

> > ### Author Response · Authors · 2019-11-15
> > **Response to AnonReviewer3 (Part 2/2)**
> >
> > > Finally, as shown in the Appendix, the method seems rather brittle, requiring both a schedule on the size of the option layer and the strength of the regularization term, something which would be difficult to do in the absence of a downstream task.
> >
> > The curriculum approach of gradually increasing the number of options has been drawn from prior work (VALOR) where the authors note that it is hard to train option discovery algorithms without such a schedule.
> > The annealing of regularization strength $\beta$ is a linear schedule common in KL-regularized objectives [2] to slowly ease in the regularization. We did not have to pick a complicated schedule for any of these. Moreover, these are design choices which are independent of the transfer performance, and are simply to maximize empowerment for a given "unsupervised" training loop. Thus, we do not find the approach to be generally brittle.
> >
> > > Similarly, I am not quite sure what to make of Figure 4 apart from the fact that different latent options yield different trajectories. See detailed comments below.
> >
> > In the MountainCar experiments, we show that in a setting where the state space (and connectivity) is more complicated than a simple 2D grid world, we are still able to obtain decision states as a sparse set of states in the environment where the agent switches between options. We show one such example of a trained model where the discovered decision states were concentrated near the x-axis (velocity = 0 line).
> >
> >
> > > * It is rather disappointing that the reverse predictor uses privileged information, in the guise of x-y features.
> >
> > Please refer to our general response about privileged (x, y) information.
> >
> > > I do not believe the sandwich bound explains why Eq. 6 helps uncover decision states. ...
> >
> > We do not claim that Eq. 6 helps discover decision states. We wrote down Eq. 6 with goal of formalizing how one might identify such decision states without explicit goals, and then observed that the objective is in fact an upper bound, which might be of more general interest. We completely agree with the reviewer that not every upper bound one could formulate would result in decision states (thus the claim is only true for the particular upper bound we derive). We are unsure what further theoretical insights we could draw from this, but would be happy to explore any concrete directions the reviewer had in mind.
> >
> > > Decision states are never clearly defined to be those having high mutual information $I(\Omega, A_t, S_t)$. It is also not clear from the main text was is being plotted in Figure 3...
> >
> > We have updated the paper to clearly define decision states and the values plotted in the heatmaps of Figure 3.
> >
> > > * Footnote 7. High variance on $\beta$=1e-4. Is it possible that too few seeds were used to estimate the standard error on the mean?
> >
> > We used the same number of random seeds (10) for all experiments in Figure 5. We did not observe a significant reduction in variance beyond 10 seeds.
> >
> > > * “Upper bound is too tight”. I don’t think this is what you mean: tight would refer to how good the upper-bound approximation is, which is different from the constraint specifying an upper-bound which is too small.
> >
> > We agree that the term tight upper bound does not refer to how close the approximation is to the true value, we have corrected this in the revision.
> >
> > > Section 4.1: Did not understand the sentence “we noticed that if an intersection is a decision state [...] having already made the decision.”
> >
> > We meant to say that not every state which looks like an intersection needs to be a decision state, if for example the part of the state space in question is only spanned by one option. We have clarified this in the revision.
> >
> > > * Notation: $p^J(s_f | \omega, s_0)$ What does J refer to? J is not defined anywhere.
> >
> > $p^J(s_f | \omega, s_t)$ is defined in Section 2.2 (VIC) as the terminal state distribution achieved when executing policy policy $\pi(a_t | \omega, s_t)$. We will clarify this better.
> >
> > > *“Decision states to be points where the cart has velocity=0”: wouldn’t this mostly be restricted to the initial state?
> >
> > The cart may come to a halt at any point on the slopes of the mountain as well, instead of just the initial state. Figure 4(d) shows the concentration of decision states near the velocity=0 line but at varying x coordinates (i.e. not just the initial x-coordinate).
> >
> > > * Section 4.1: Did not understand what is meant by “where trajectories associated with different options intersect”? What does this mean concretely in MountainCar for trajectories to intersect?
> >
> > Trajectories in MountainCar are visualized in the position-velocity space and if trajectories corresponding to two different options reach the same position and velocity, then they are said to intersect.
> >
> > [2] Higgins, Irina, et al. "beta-VAE: Learning Basic Visual Concepts with a Constrained Variational Framework." ICLR 2.5 (2017): 6.

---

### Official Review · AnonReviewer4 · 2019-11-26
**Official Blind Review #4**

**Rating:** 3

**Review:**

This paper proposes a mechanism for identifying decision states, even on previously unseen tasks. Decision states are states from which the option taken has high mutual information with the final state of that option, but low mutual information with the action at a time-step, given the current state. An intrinsic reward based on an upper bound of the relevant mutual information speeds up learning in similar environments that the agent has not encountered.

A key contribution of this work is extending the notion of goal-driven decision states to goal-independent decision states. The authors also introduce an interesting upper bound on the mutual information between options and final states.

The authors provide an empirical evaluation that supports their central claims.

I recommend this paper be accepted because it contributes an interesting theoretical result, a definition of decision state that does not depend on extrinsic rewards, and an algorithm to find such decision states.

Further suggestions to improve clarity that did not influence the decision:
* The acronym VIC is used frequently throughout the introduction, but is not explained until section 2. Please introduce variational intrinsic control in the introduction.
* The partial observability claim is not substantiated by the experiments. From the general response to reviewers, "Therefore, we make the assumption that the complete state is available, in order to study unsupervised decision states." It is not a problem to assume that the complete state is available, but claiming to generalize to partial observability is not entirely correct, even if your method handles the same semi-partially observable case as previous work like VIC or DIAYN.
* Please explicitly describe the motivation for using a bottleneck variable.
* In MDPs with fixed episode length, the probability of termination is only non-zero on the last time step. Therefore, information about the time step must be included in a Markov state. The options in this paper terminate based on a time horizon, but there is no mention of whether the intra-option time step is included in the state or bottleneck variables.

**Experience Assessment:**

I do not know much about this area.

**Review Assessment: Checking Correctness Of Derivations And Theory:**

I assessed the sensibility of the derivations and theory.

**Review Assessment: Checking Correctness Of Experiments:**

I assessed the sensibility of the experiments.

**Review Assessment: Thoroughness In Paper Reading:**

I read the paper at least twice and used my best judgement in assessing the paper.

---

### Author Response · Authors · 2019-11-15
**General Response to All Reviewers**

We thank the reviewers for the detailed feedback and are encouraged that they thought the paper provides interesting theoretical results [R3], and is well motivated and interesting [R2, R1].

We have also uploaded a revision addressing several suggestions. Below, we address general concerns and individually reply to each reviewer.

We would like to reiterate that our key contribution is in providing a principled mathematical framework to identify decision states in goal-free settings. Towards this, we derive an upper bound on the VIC objective [R3] and demonstrate that the resulting objective is useful for identifying decision states. Next, we show that incentivizing decision state visitation leads to better transfer performance. Our paper focuses on replicating the exact settings and experiments used by InfoBot [R3]. Furthermore, while methodologically related, we believe direct comparisons to experiments in DIAYN and VALOR (which focus more on skill learning as opposed to decision states) are not central to the goal of the paper.

[R2, R1] Alignment to human intuition: Although our decision states align occasionally with human intuition, we are ultimately interested in transfer to goal-driven tasks. While interpretable decision states do emerge in InfoBot with goal conditioning, our formulation chooses decision states which have high option-information (emerging naturally from our mutual information computation in Eq. 8 and Appendix A5), as opposed to states which have high goal-information (for a particular goal, as in InfoBot). Because of this averaging, our unsupervised decision states are not expected to be as interpretable as InfoBot.

However, if we condition on a single option (a loose proxy for a goal in our setting), the decision states become more interpretable and evident. However, we refrain from making strong claims about them being exactly human interpretable. Moreover, not every state which looks like an intersection needs to be a decision state, if for example the part of the state space in question is only spanned by one option.

Ultimately, we take the view that a useful decision state is one that is useful for a downstream task. Thus, we don't seek a binary threshold for decision states, but view them as part of a continuum.

[R3] Optimal value of $\beta$: The choice of $\beta$ is inherent in unsupervised representation learning is no different from say, choosing the number of clusters in K-means. Picking $\beta$ does not make our method supervised, and should instead be thought of as model selection. Further, our experiments provide initial evidence that the same value of $\beta$ works well across multiple downstream transfer environments.

[R2, R3] We provide further justifications for why the use of privileged (x, y) information in the MultiRoom environments does not render our experimental results less meaningful:

1. The (x, y) coordinate information is available to all baselines, including ones which infer options in during training [R2] (DIAYN) or condition on a goal vector (InfoBot — uses relative coordinates of the goal w.r.t. the current state). Considering choices made in prior work, our usage of (x, y) coordinates is fair. As explained in 3), to the best of our knowledge, it is an open (and orthogonal) problem to understand how to use VIC/DIAYN line of work with only partial state observations and learn explicit options.
2. No global (x, y) coordinates are used when providing the bonus at transfer time (which is what we ultimately care about). Transfer only uses the state encoder and the policy (and not the option inference network) — in a novel environment, one does not need global (x, y) coordinates; only the trained encoder is used to provide an exploration bonus from partial observations.
3. Appendix (A6): all option-based methods (DS-VIC, DIAYN) use the MultiRoom environment in which the layout of the rooms and doors is fixed for discovering options in the pre-training phase. In some sense, it is reasonable to assume that the agent (over multiple episodes of training) is building an internal map and estimating its state in the map using a black box SLAM module. Integrating state estimation and mapping into a deep learning pipeline is an active area of research that is orthogonal to our primary contribution [1]. Therefore, we make the assumption that the complete state is available, in order to study unsupervised decision states.

We also discuss in Section A6 about our attempts to discover options without the X-Y coordinates in the pre-training phase.

[R1, R2, R3] Missing baseline of VIC + Max-Entropy, $\beta = 0$: We ran an additional experiment with $\beta=0$, and found performance close to $\beta=10^-6$. We have updated Fig. 5 with this. Mathematically, this baseline is identical to running VIC with max-entropy.

References
[1]: Gupta, Saurabh et. al. 2017. “Cognitive Mapping and Planning for Visual Navigation.”

---

### Decision · Program_Chairs · 2019-12-19

**Decision:**

Reject

**Comment:**

This work is interesting because it's aim is to push the work in intrinsic motivation towards crisp definitions, and thus reads like an algorithmic paper rather than yet another reward heuristic and system building paper. There is some nice theory here, integration with options, and clear connections to existing work.

However, the paper is not ready for publication. There were were several issues that could not be resolved in the reviewers minds (even after the author response and extensive discussion). The primary issues were: (1) There was significant confusion around the beta sensitivity---figs 6,7,8 appear misleading or at least contradictory to the message of the paper. (2) The need for x,y env states. (3) The several reviewers found the decision states unintuitive and confused the quantitative analysis focus if they given the authors primary focus is transfer performance. (4) All reviewers found the experiments lacking. Overall, the results generally don't support the claims of the paper, and there are too many missing details and odd empirical choices.

Again, there was extensive discussion because all agreed this is an interesting line of work. Taking the reviewers excellent suggestions on board will almost certainly result in an excellent paper. Keep going!